# Traceability Research on *Dendrobium devonianum* Based on SWATHtoMRM

**DOI:** 10.3390/foods12193608

**Published:** 2023-09-28

**Authors:** Tao Lin, Xinglian Chen, Lijuan Du, Jing Wang, Zhengxu Hu, Long Cheng, Zhenhuan Liu, Hongcheng Liu

**Affiliations:** 1Quality Standards and Testing Technology Research Institute, Yunnan Academy of Agricultural Sciences, Kunming 650205, China; lintaonj@126.com (T.L.); chen544141152@163.com (X.C.); 15825298061@163.com (L.D.); lzh@yaas.org.cn (Z.L.); 2Longling Agricultural Environmental Protection Monitoring Station, Baoshan 678300, China; jingjingmichuer@163.com (J.W.); 18708755021@139.com (Z.H.); 3SCIEX Analytical Instrument Trading Co., Ltd., Shanghai 200335, China; long.cheng@sciex.com

**Keywords:** time-of-flight mass spectrometry, identification, *Dendrobium devonianum*, Longling area

## Abstract

SWATHtoMRM technology was used in this experiment to further identify and trace the sources of *Dendrobium devonianum* and *Dendrobium officinale* produced in the same area using TOF and MS-MRM. After the conversion of the R package of SWATHtoMRM, 191 MRM pairs of positive ions and 96 pairs of negative ions were obtained. *Dendrobium devonianum* and *Dendrobium officinale* can be separated very well using the PCA and PLS-DA analysis of MRM ion pairs; this shows that there are obvious differences in chemical composition between *Dendrobium devonianum* and *Dendrobium officinale*, which clearly proves that the pseudotargeted metabolomics method based on SWATHtoMRM can be used for traceability identification research. A total of 146 characteristic compounds were obtained, with 20 characteristic compounds in *Dendrobium devonianum*. The enrichment pathways of the characteristic compounds were mainly concentrated in lipids and atherosclerosis, chagas disease, fluid shear stress and atherosclerosis, proteoglycans in cancer, the IL-17 signaling pathway, the sphingolipid signaling pathway, diabetic cardiomyopathy, arginine and proline metabolism, etc., among which the lipid and atherosclerosis pathways were more enriched, and 11 characteristic compounds affected the expression levels of IL-1, TNFα, CD36, IL-1β, etc. These can be used as a reference for research on variety improvement and active substance accumulation in *Dendrobium devonianum* and *Dendrobium officinale*.

## 1. Introduction

*Dendrobium devonianum* is a characteristic Chinese herbal medicine produced in the Longling area of Yunnan, China, and it is also a Chinese plant that can be eaten as food [1,2,3]. *Dendrobium devonianum* has good biological health effects [4,5], and the geographical location where it grows has a large impact on it. Among them, the Longling area produces nationally important products, but the *Dendrobium devonianum* that grows in other parts of Yunnan is not a notable product [6]. In order to counterfeit *Dendrobium devonianum* produced in the Longling area of Yunnan, *Dendrobium officinale* is often planted in the Longling area and sold as *Dendrobium devonianum*. Although the appearance and shape of *Dendrobium devonianum* and *Dendrobium officinale* produced in the Longling area are different, it is difficult to identify when it is dried or crushed into a powder, which seriously affects the quality evaluation and origin traceability of *Dendrobium devonianum* [7]. Therefore, it is essential to establish an effective *Dendrobium devonianum* traceability technology.

High-resolution mass spectrometry is one of the commonly used and effective traceability technologies [8,9,10,11]. In our laboratory, TOF and UPLC-PDA have also been used to trace the origins of *Dendrobium devonianum* and *Dendrobium officinale* produced in the same area; *Dendrobium devonianum* and *Dendrobium officinale* can be better distinguished through relevant PCA and VIP analyses, etc., but the number of confirmed differential markers obtained was small, and the amount of characteristic information about the differential markers was low [7,12]. This also shows that, although they have a wide coverage, TOF or UPLC-PDA have a dynamic range, quantitative accuracy, and significantly reduced sensitivity, and the final characteristic information total obtained is less [13,14,15].

With the emergence of sequential windowed acquisition of all theoretical fragment ions to multiple reaction monitoring (SWATHtoMRM) technology, the problems of the low sensitivity of mass spectrometry in full scan mode and accuracy and reliability in the process of structure identification have been effectively solved. Through a non-targeted analysis of SWATH data, the generation of MRM ion pair information, and the targeted analysis of each correlated MRM ion pair, SWATHtoMRM technology combines the powerful qualitative ability of SWATH technology with the precise quantitative ability of MRM technology to achieve high coverage and accurate quantification of known and unknown metabolites detected in a non-targeted analysis [16]. At present, this has been widely used in metabolomics, foodomics, etc. [17,18,19].

In order to obtain as much information as possible about the characteristic compounds in *Dendrobium devonianum* produced in the Longling area of Yunnan, SWATHtoMRM technology was used in this experiment to further identify and trace the sources of *Dendrobium devonianum* and *Dendrobium officinale* produced in the same area using TOF and MS-MRM to analyze the known and unknown metabolites in *Dendrobium devonianum* and *Dendrobium officinale* to obtain more information on the characteristic metabolites in *Dendrobium devonianum*.

## 2. Materials and Methods

### 2.1. Sample Collection and Preparation

Twenty-six samples of *Dendrobium devonianum* and *Dendrobium officinale*, thirteen each, were collected from the Longling area of Yunnan, China, in 2020. Each collected sample was composed of 10 fresh branches, kept under the same growth condition. The branches were cut into lengths of about 5 cm, dried at 60 °C, crushed at a high speed, passed through a 0.28 μm sample sieve, and stored in the laboratory at 4 °C in the dark.

### 2.2. Chemicals and Reagents

HPLC-grade acetonitrile, isopropanol, and methanol were purchased from Merck (Darmstadt, Germany). HPLC-grade ammonium acetate and formic acid were purchased from DiKMA Technologies (Beijing, China). Ultrapure water was prepared using Elga’s water system (Wycombe, UK).

### 2.3. Sample Preparation and Analysis

#### 2.3.1. Sample Preparation and Instrumental Method

First, 2 g of sample was placed into a 50 mL centrifuge tube, 20 mL of methanol–water solution (V:V = 90:10) was added and vortexed for 1 min, then ultrasonic was extracted for 30 min and centrifuged at 5000 r/min for 5 min, and the supernatant was filtered through a 0.22 μm filter membrane.

The SCIEX X500R QTOF system (Framingham, MS, USA) used was equipped with an ExionLC AD ultra-high-performance liquid chromatography (Framingham, MS, USA) and Waters ACQUITY UPLC BEH C18 column (2.1 × 100 mm, 1.7 μm, Waters, Milford, MA, USA). Referring to the relevant parameters in reference [7]: solvent A was 2 mM ammonium acetate in ultrapure water with 0.01% formic acid, and B was the mixed solution of acetonitrile, isopropanol, and water (V:V:V = 47.5:47.5:5) containing 2 mM formic acid and 0.01% formic acid. The flow rate for UPLC was 0.4 mL/min with the following gradients: 10% B (0~5.0 min),  10% B~50% B (5.0~6.0 min),  50% B~95% B (6.0~15.0 min), 95% B~100% B (15.0~20.0 min), 100% B (20.0~35.0 min), 100% B~5% B (35.0~35.1 min), and 5% B (35.1~40.0 min). The injection volume was 5 µL. Data were collected using primary and secondary mass spectrometry, among which the MS scan range was 100~1500 *m*/*z*, and the MS IDA scan range was 50~1500 *m*/*z*, CE = ±30 V.

#### 2.3.2. MRM Data Collection

The wiff format of the sample data collected via TOF was converted to the mzXML format using MSConvert software (3.0.4140), and the R package (4.3.1) of SWATHtoMRM was used for the conversion of MRM transitions. Twenty-six samples were analyzed using the AB SCIEX 4500 system (Framingham, MS, USA). The same chromatographic column and gradient elution conditions as in QTOF-MS data collection were used, along with the converted MRM transition and schedule mode (MRM detection window: 50 s), where the DP was uniformly ±50 V and the collision energy was uniformly ±40 V. At the same time, the same volume of the extraction solution of each sample in this experiment was drawn and mixed to make a quality control sample (QC). During the sample analysis process, the analysis of QC samples was performed after every 5 samples to monitor the sensitivity and stability of the instrument and to perform a subsequent data analysis and correction.

#### 2.3.3. Data Processing and Statistical Analysis

The ions whose response peak area was lower than 10^2^ were deleted, and the ion pairs with a high response intensity were retained, which were analyzed using the peak area response of each ion pair of the QC sample. Lists of peak areas corresponding to MRM were imported into MetaboAnalyst (https://www.metaboanalyst.ca/ (accessed on 7 July 2023)) for a principal component analysis (PCA), ANOVA, false discovery rate (FDR), and VIP analysis to find differential MRM ion pairs. Differential MRM ion pairs were compared to TOF data using ion pair information and the peak time to obtain primary and secondary mass spectrum information of differential compounds, and the chemical structure was identified using the Natural Product s-TCM Library_1.0 established by SCIEX and the online ChemSpider database (HMDB, Massbank, Pubmed, etc.). In order to further analyze the possible signaling pathways affecting differential metabolites, the differential metabolites were imported into the MetaboAnalyst5.0 (https://www.metaboanalyst.ca/ (accessed on 7 July 2023)) online website to analyze the main enriched KEGG biosynthetic pathways.

## 3. Results

### 3.1. Analytical Characteristics of SWATHtoMRM Method

After the conversion of the R package of SWATHtoMRM, the ion pairs were obtained, in which the positive ions totaled 2439 pairs of MRMs and the negative ions totaled 601 pairs of MRMs (Appendix A). From the *m*/*z* distribution of the parent ions in Figure 1, it can be seen that the MRM of the positive ions was mainly concentrated in the range of 300–600. After screening the ion pairs with the low-level response QC samples, 191 MRM pairs of positive ions and 96 pairs of negative ions were obtained (Appendix A).

### 3.2. MRM Data Analysis

As can be seen from Figure 2, the MRM ion pairs obtained after screening were analyzed using a PCA, and the degree of polymerization of the QC was high, indicating that the data were stable and the quality was guaranteed. It can be seen that *Dendrobium devonianum* and *Dendrobium officinale* can be separated very well using a PCA analysis; this shows that there are obvious differences in the chemical composition between *Dendrobium devonianum* and *Dendrobium officinale*.

The difference in the comparisons in this experiment is mainly reflected in the difference in the peak areas of the common components contained in *Dendrobium devonianum* and *Dendrobium officinale*. The two principal components, PC1 and PC2, accounted for 27.0% and 23.8% of the total difference, respectively, indicating that by comparing the difference in the peak area of the common components in *Dendrobium devonianum* and *Dendrobium officinale*, it is possible to effectively distinguish *Dendrobium devonianum* and *Dendrobium officinale*. It also shows that the ion pairs converted using SWATHtoMRM were analyzed via PCA, and the two *Dendrobium* samples were densely gathered together, which clearly proves that the pseudotargeted metabolomics method based on SWATHtoMRM can be used for traceability identification research.

On the other hand, PLS-DA was also used for the analysis, and the analysis results were consistent with the PCA, and *Dendrobium devonianum* and *Dendrobium officinale* can be separated well (Figure 3), as can be seen in the PLS-DA cross-validation data (Figure 4). R2 is the correlation coefficient of cross validation, and the values of components of 1–5 were 0.96224, 0.98362, 0.99473, 0.99767, and 0.99929, respectively, which were close to 1, indicating that their fitting degree was good. Q2 represents the predictive performance of the PLS-DA model, and Q2 was higher than 0.9, so it can be considered a very good model in this experiment.

### 3.3. ANOVA, FDR, and VIP Analysis

ANOVA, FDR, and VIP analyses were performed on the obtained MRM ion pairs, and the MRM ion pairs with a *p*-value < 0.01, FDR < 0.05, and VIP > 1 were selected as the ion pairs with large differences. A total of 146 characteristic compounds were obtained (Appendix A).

As shown in Figure 5 and Figure 6, the retention time of the 146 characteristic compounds was mainly the range of 6–20 min, and the molecular weight was mainly concentrated between 200 and 300 and 500 and 700. According to the chromatographic conditions in Section 2.3.1, during a time period of 6–15 min, the organic phase was from 50 to 95%, and at 15–20 min, the organic phase was from 95 to 100%. The main elution components were medium and medium-to-small polar compounds, which were the same as the main components in *Dendrobium*, which were consistent with alkaloids, flavonoids, phenanthrenes, and bibenzyls [20].

It can be seen from the analysis of the VIP scores (Figure 7), volcano map (Figure 8), and heat map (Figure 9) of the 50 compounds with large differences that, as shown in Table 1, there were 20 characteristic compounds in *Dendrobium devonianum*, the content of *Dendrobium devonianum* was larger than that of *Dendrobium officinale*, and the content of 30 characteristic compounds was smaller than that of *Dendrobium officinale*. The difference in the contents of the common compounds was the largest (*p* = 6.32 × 10^−17^), which may be the characteristic component of *Dendrobium devonianum*; the three differential compounds with a reduced content may be the characteristic components of *Dendrobium officinale*. As shown in Figure 10, the contents of the five characteristic compounds of *Dendrobium devonianum* and *Dendrobium officinale* were very different. The normalized concentrations of the five characteristic compounds were close to +1 and −1, respectively, and the difference can be clearly seen after normalization.

### 3.4. Structural Identification of Characteristic Compounds

Using the Natural Products s-TCM Library_1.0 and online ChemSpider database, a total of 34 characteristic compounds were identified, including 20 in the positive ion mode and 14 in the negative ion mode, as shown in Table 2.

### 3.5. KEGG Pathway Analysis of Dendrobium devonianum and Dendrobium officinale

According to the 11 of the top 50 characteristic compounds with a confirmed chemical structure obtained above, a KEGG pathway analysis was performed, and the top 20 pathways with *p* ≤ 0.05 were selected for visual depiction. As shown in Figure 11, the enrichment pathways of the characteristic compounds were mainly concentrated in the lipids and atherosclerosis, chagas disease, fluid shear stress and atherosclerosis, proteoglycans in cancer, IL-17 signaling pathway, sphingolipid signaling pathway, diabetic cardiomyopathy, arginine and proline metabolism, etc., among which the lipid and atherosclerosis pathways were more enriched and 11 characteristic compounds could better affect the expression levels of IL-1, TNFα, CD36, IL-1β, etc. (Figure 12). On the other hand, this also proved that the metabolic processes of lipids and atherosclerosis can be better regulated by *Dendrobium devonianum*, which is consistent with the biological health effects of *Dendrobium nobile* reported in the literature [67,68,69], which can be used as a reference for future research on variety improvement and active substance accumulation in *Dendrobium devonianum* and *Dendrobium officinale*.

## 4. Conclusions

In this study, SWATHtoMRM technology was used in this experiment to further identify and trace the sources of *Dendrobium devonianum* and *Dendrobium officinale* produced in the same area using TOF and MS-MRM. After the conversion of the R package of SWATHtoMRM, the ion pairs were obtained, in which the positive ions totaled 2439 pairs of MRMs, and the negative ions totaled 601 pairs of MRMs. After screening the ion pairs with low-level response QC samples, 191 MRM pairs of positive ions and 96 pairs of negative ions were obtained. *Dendrobium devonianum* and *Dendrobium officinale* can be separated very well via a PCA analysis of MRM ion pairs; this shows that there are obvious differences in chemical composition between *Dendrobium devonianum* and *Dendrobium officinale*. The difference in the comparisons in this experiment mainly reflect the differences in the peak areas of the common components contained in *Dendrobium devonianum* and *Dendrobium officinale*. This also shows that the ion pairs converted using SWATHtoMRM were analyzed via PCA, and the two *Dendrobium* samples were densely gathered together, which clearly proves that the pseudotargeted metabolomics method based on SWATHtoMRM can be used for traceability identification research. On the other hand, *Dendrobium devonianum* and *Dendrobium officinale* can be separated well via PLS-DA, as can be seen through PLS-DA cross validation. The R2 values of components 1–5 were 0.96224, 0.98362, 0.99473, 0.99767, and 0.99929, respectively, which were close to 1, indicating that their fitting degree was good, and the Q2 was above 0.9, which indicates a very good model.

The ANOVA FDR and VIP analyses were performed on the obtained MRM ion pairs. A total of 146 characteristic compounds were obtained. There were 20 characteristic compounds in *Dendrobium devonianum*; the content of *Dendrobium devonianum* was larger than that of *Dendrobium officinale*; and the content of 30 characteristic compounds was smaller than that of *Dendrobium officinale*. The difference in the contents of the most common compounds was the largest (*p* = 6.32 × 10^−17^), which may represent the characteristic component of *Dendrobium devonianum*; three differential compounds with reduced contents may be the characteristic components of *Dendrobium officinale*. The enrichment pathways of the characteristic compounds were mainly concentrated in the lipids and atherosclerosis, chagas disease, fluid shear stress and atherosclerosis, proteoglycans in cancer, IL-17 signaling pathway, sphingolipid signaling pathway, diabetic cardiomyopathy, arginine and proline metabolism, etc., among which the lipid and atherosclerosis pathways were more enriched and 11 characteristic compounds could better affect the expression levels of IL-1, TNFα, CD36, IL-1β, etc., which can be used as a reference for future research on variety improvement and active substance accumulation in *Dendrobium devonianum* and *Dendrobium officinale*.

## Figures and Tables

**Figure 1 foods-12-03608-f001:**
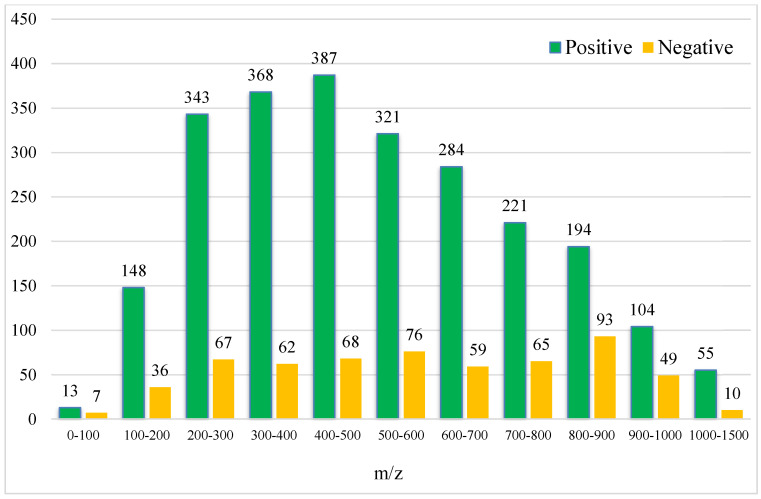
Molecular weight distribution of compounds converted using SWATHtoMRM.

**Figure 2 foods-12-03608-f002:**
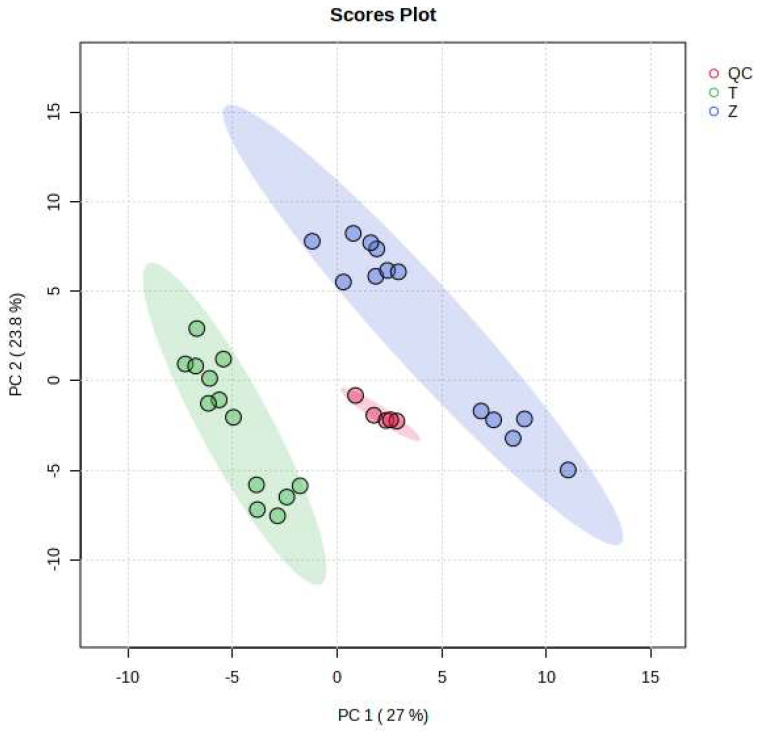
Loading plots of PCA of *Dendrobium devonianum* and *Dendrobium officinale* (T: *Dendrobium officinale*, Z: *Dendrobium devonianum*, QC: QC samples).

**Figure 3 foods-12-03608-f003:**
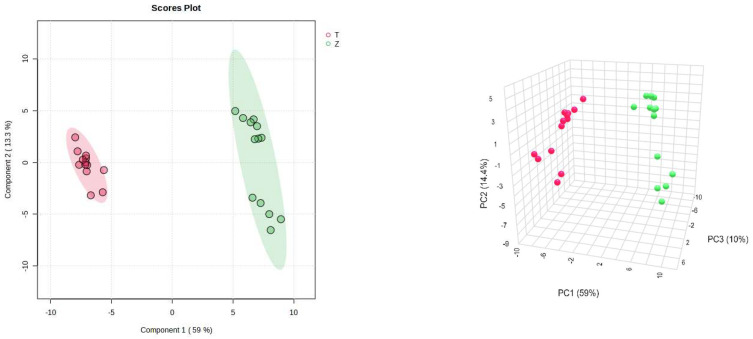
PLS-DA score chart and model diagram of *Dendrobium devonianum* and *Dendrobium officinale* (T: *Dendrobium officinale*, red range; Z: *Dendrobium devonianum*, green range).

**Figure 4 foods-12-03608-f004:**
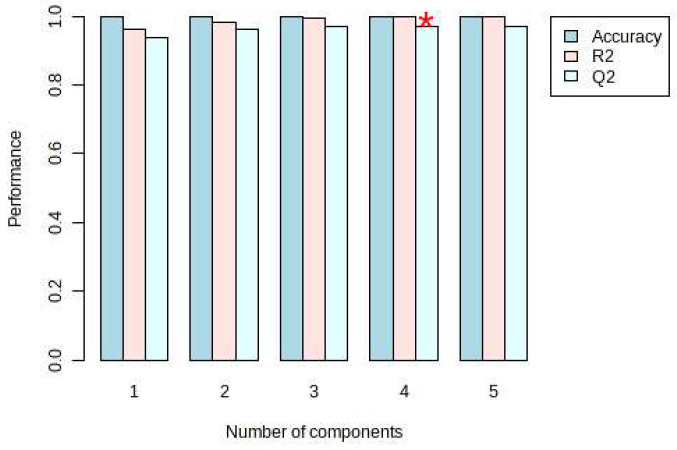
PLS-DA cross validation details of *Dendrobium devonianum* and *Dendrobium officinale*. (* The Q2 value was 0.96961, which was also the largest).

**Figure 5 foods-12-03608-f005:**
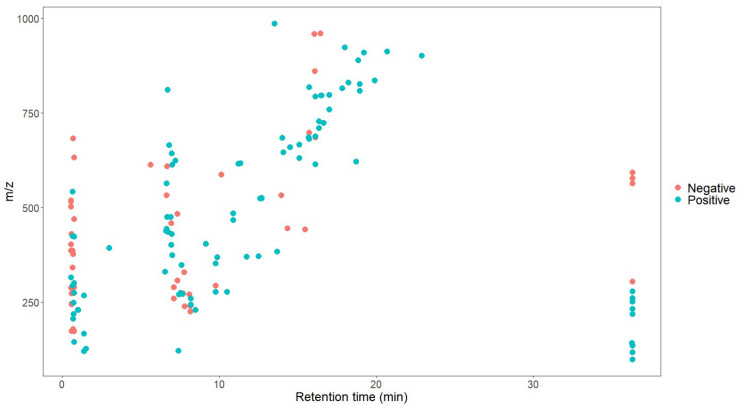
Scatter plot of 146 characteristic compounds.

**Figure 6 foods-12-03608-f006:**
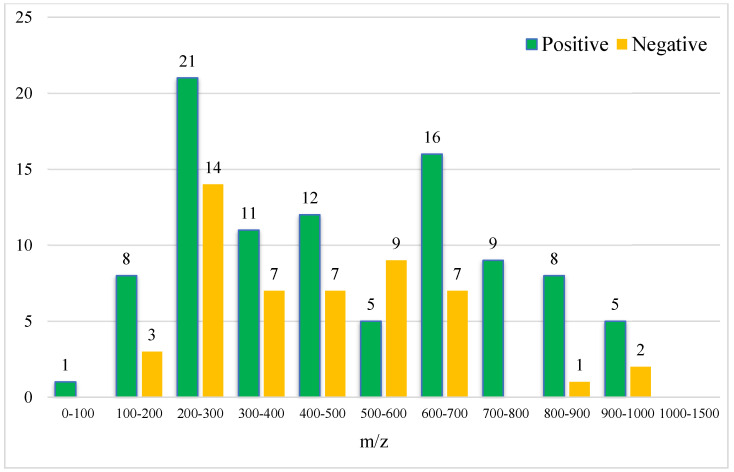
Molecular weight distribution of 146 characteristic compounds (the numbers on the bar graph represent the number of characteristic compounds in that molecular weight range).

**Figure 7 foods-12-03608-f007:**
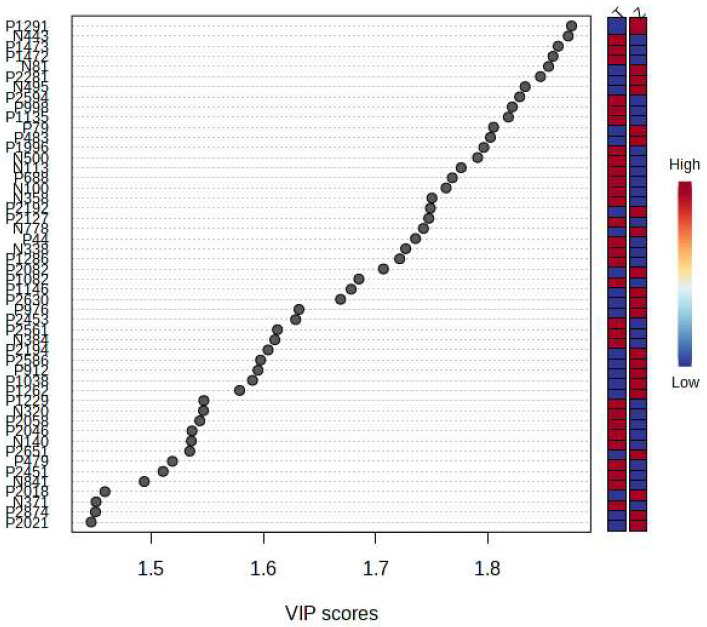
The VIP scores of *Dendrobium devonianum* and *Dendrobium officinale* (T: *Dendrobium officinale*, Z: *Dendrobium devonianum*).

**Figure 8 foods-12-03608-f008:**
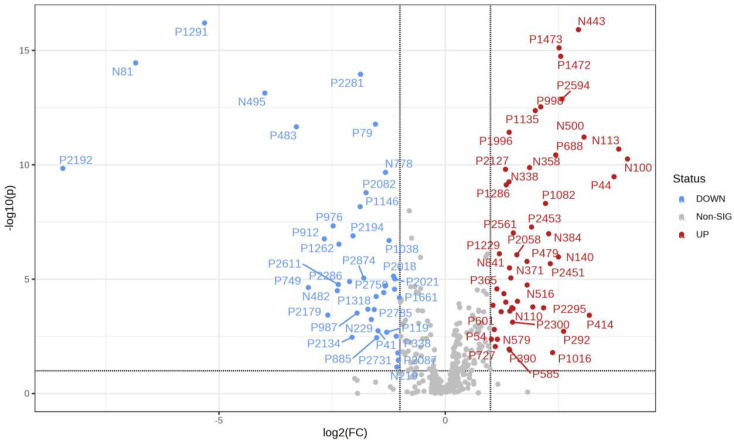
A volcano plot of *Dendrobium devonianum* and *Dendrobium officinale*.

**Figure 9 foods-12-03608-f009:**
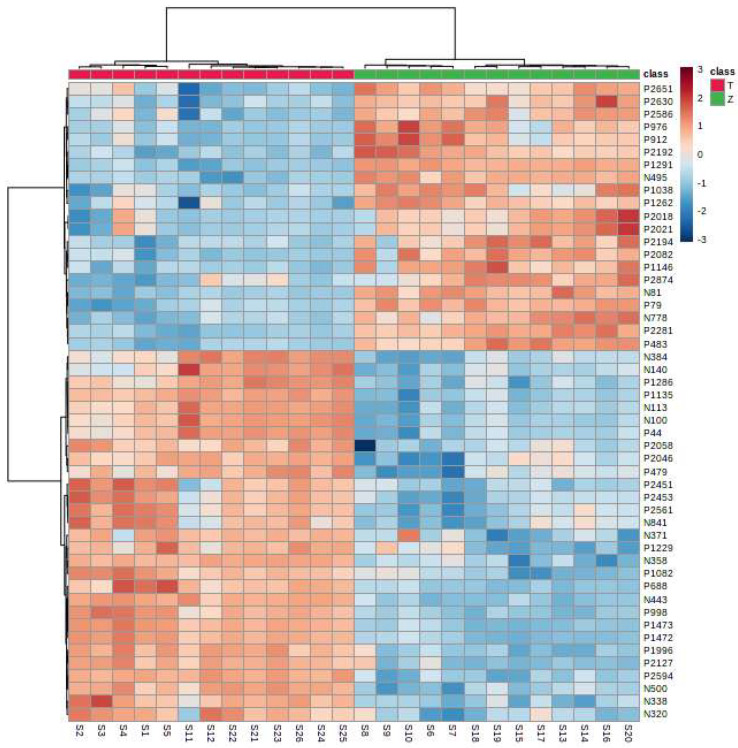
A heatmap of *Dendrobium devonianum* (Z) and *Dendrobium officinale* (T).

**Figure 10 foods-12-03608-f010:**
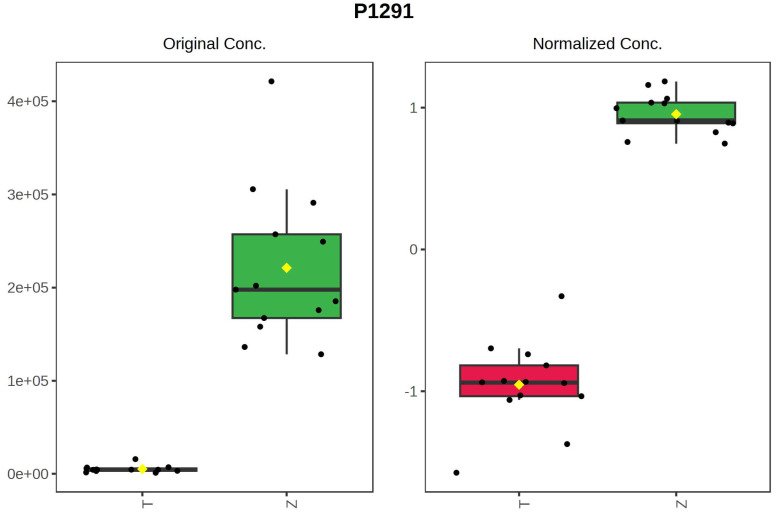
Box plot of the five characteristic compounds (T: *Dendrobium officinale*, red range, Z: *Dendrobium devonianum*, green range).

**Figure 11 foods-12-03608-f011:**
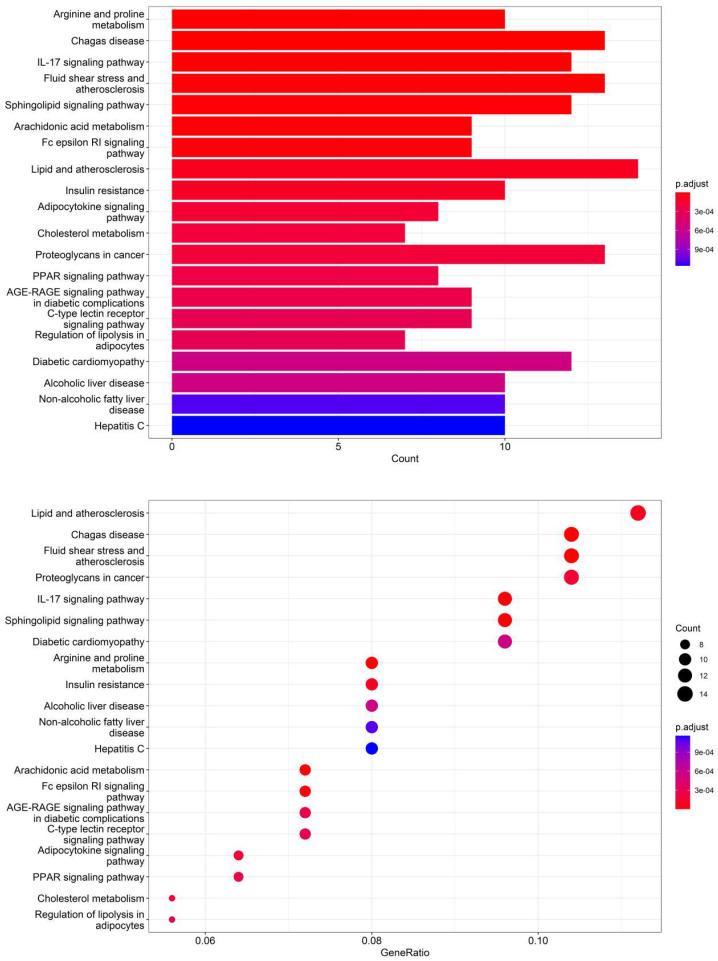
Barplot and dotplot of the top 20 KEGG enrichment pathways based on 11 characteristic compounds.

**Figure 12 foods-12-03608-f012:**
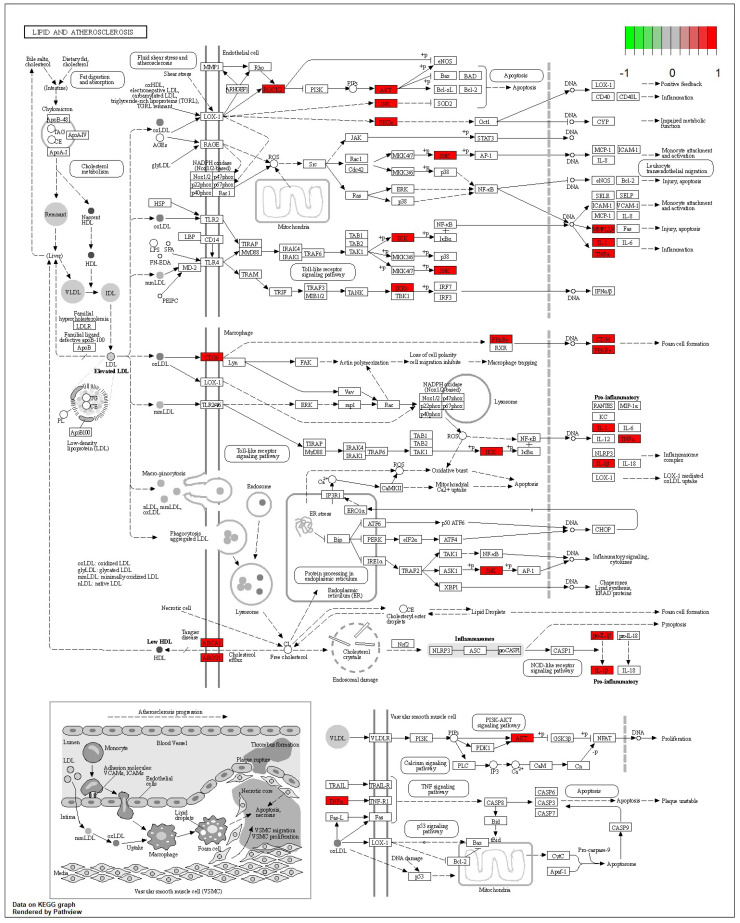
Lipid and atherosclerosis metabolic pathway.

**Table 1 foods-12-03608-t001:** Information on characteristic compounds.

Compound Code	Precursor Ion (*m*/*z*)	Product Ion (*m*/*z*)	Retention Time (min)	*p* Value	FDR	VIP	Changes in Content	
							*Dendrobium devonianum*	*Dendrobium officinale*
P1291	438.2390	70.0651	6.60	6.32 × 10^−17^	2.20 × 10^−14^	1.8749	↑	↓
N443	533.1296	124.9909	6.64	1.24 × 10^−16^	2.20 × 10^−14^	1.8719	↓	↑
P1473	476.2134	137.0597	6.90	7.73 × 10^−16^	9.17 × 10^−14^	1.8630	↓	↑
P1472	476.1928	137.0236	6.67	1.81 × 10^−15^	1.61 × 10^−13^	1.8583	↓	↑
N81	239.0707	196.0524	7.79	3.52 × 10^−15^	2.50 × 10^−13^	1.8544	↓	↑
P2281	682.5257	248.2375	15.73	1.11 × 10^−14^	6.57 × 10^−13^	1.8472	↑	↓
N495	609.1449	311.0557	6.66	7.31 × 10^−14^	3.72 × 10^−12^	1.8335	↑	↓
P2594	811.6109	235.0970	6.69	1.34 × 10^−13^	5.97 × 10^−12^	1.8287	↓	↑
P998	374.1813	154.0624	7.00	2.92 × 10^−13^	1.16 × 10^−11^	1.8220	↓	↑
P1135	401.1601	167.0706	6.93	4.30 × 10^−13^	1.53 × 10^−11^	1.8185	↓	↑
P79	144.1019	70.0650	0.73	1.68 × 10^−12^	5.43 × 10^−11^	1.8053	↑	↓
P483	272.2222	105.0698	7.66	2.19 × 10^−12^	6.49 × 10^−11^	1.8026	↑	↓
P1996	613.2288	167.0703	6.99	3.79 × 10^−12^	1.04 × 10^−10^	1.7966	↓	↑
N500	614.2169	61.9883	5.61	6.18 × 10^−12^	1.57 × 10^−10^	1.7911	↓	↑
N113	273.1128	137.0238	7.68	2.06 × 10^−11^	4.89 × 10^−10^	1.7764	↓	↑
P688	315.9915	119.0186	0.56	3.73 × 10^−11^	8.29 × 10^−10^	1.7686	↓	↑
N100	271.0969	213.0553	8.07	5.57 × 10^−11^	1.17 × 10^−9^	1.7630	↓	↑
N358	459.1506	149.0598	6.94	1.32 × 10^−10^	2.61 × 10^−9^	1.7504	↓	↑
P2192	665.1730	271.0600	6.80	1.44 × 10^−10^	2.71 × 10^−9^	1.7490	↑	↓
P2127	643.2394	167.0702	6.97	1.58 × 10^−10^	2.82 × 10^−9^	1.7476	↓	↑
N778	861.5465	279.2321	16.08	2.15 × 10^−10^	3.64 × 10^−9^	1.7429	↑	↓
P44	121.0647	78.0463	7.39	3.33 × 10^−10^	5.39 × 10^−9^	1.7357	↓	↑
N338	443.3365	309.3152	15.45	5.58 × 10^−10^	8.63 × 10^−9^	1.7270	↓	↑
P1286	436.2185	219.0805	6.72	7.55 × 10^−10^	1.12 × 10^−8^	1.7217	↓	↑
P2082	631.4929	81.0186	15.11	1.66 × 10^−9^	2.36 × 10^−8^	1.7072	↑	↓
P1082	393.2233	107.0491	3.00	4.90 × 10^−9^	6.71 × 10^−8^	1.6853	↓	↑
P1146	404.1015	93.0698	9.13	6.77 × 10^−9^	8.93 × 10^−8^	1.6783	↑	↓
P2630	826.6767	262.2528	18.95	1.03 × 10^−8^	1.31 × 10^−7^	1.6689	↑	↓
P976	369.2637	277.2160	9.85	4.70 × 10^−8^	5.77 × 10^−7^	1.6318	↑	↓
P2453	760.5814	299.0617	17.01	5.26 × 10^−8^	6.24 × 10^−7^	1.6289	↓	↑
P2561	798.6059	601.5166	17.00	9.51 × 10^−8^	1.09 × 10^−6^	1.6126	↓	↑
N384	483.1995	134.0368	7.33	1.03 × 10^−7^	1.15 × 10^−6^	1.6103	↓	↑
P2194	666.5303	531.4039	15.11	1.27 × 10^−7^	1.37 × 10^−6^	1.6043	↑	↓
P2586	808.6658	262.2528	18.95	1.60 × 10^−7^	1.68 × 10^−6^	1.5976	↑	↓
P912	353.2686	93.0697	9.75	1.73 × 10^−7^	1.76 × 10^−6^	1.5953	↑	↓
P1038	384.3474	69.0697	13.67	2.04 × 10^−7^	2.01 × 10^−6^	1.5904	↑	↓
P1262	430.1715	145.0284	6.97	2.95 × 10^−7^	2.84 × 10^−6^	1.5789	↑	↓
P1229	425.1161	245.0510	0.65	7.76 × 10^−7^	7.14 × 10^−6^	1.5470	↓	↑
N320	430.9467	114.9882	0.57	7.83 × 10^−7^	7.14 × 10^−6^	1.5467	↓	↑
P2058	625.2548	421.1464	7.20	8.59 × 10^−7^	7.64 × 10^−6^	1.5434	↓	↑
P2046	621.5454	147.1170	18.71	1.05 × 10^−6^	9.01 × 10^−6^	1.5364	↓	↑
N140	289.1076	137.0239	7.10	1.06 × 10^−6^	9.01 × 10^−6^	1.5359	↓	↑
P2651	836.6972	262.2531	19.89	1.11 × 10^−6^	9.16 × 10^−6^	1.5344	↑	↓
P479	271.0966	182.0728	7.44	1.68 × 10^−6^	1.36 × 10^−5^	1.5190	↓	↑
P2451	760.5056	299.0617	17.01	2.10 × 10^−6^	1.66 × 10^−5^	1.5106	↓	↑
N841	961.6075	112.9853	16.45	3.22 × 10^−6^	2.49 × 10^−5^	1.4939	↓	↑
P2018	616.3460	313.2734	11.22	7.40 × 10^−6^	5.61 × 10^−5^	1.4588	↑	↓
N371	470.1507	128.0353	0.75	8.85 × 10^−6^	6.51 × 10^−5^	1.4509	↓	↑
P2874	986.6047	611.4668	13.52	8.96 × 10^−6^	6.51 × 10^−5^	1.4504	↑	↓
P2021	617.3496	313.2734	11.35	9.75 × 10^−6^	6.95 × 10^−5^	1.4465	↑	↓

P: positive; N: negative; ↑: content went up; ↓: content went down.

**Table 2 foods-12-03608-t002:** Information on characteristic compounds identified.

Compound Code	Compound Name	Molecular Formula	Adduct Ion	Mass Error (ppm)	References
P44 *	4-Hydroxybenzoic acid	C_7_H_6_O_3_	[M − H_2_O + H]^+^	0.8	[21]
P79 *	Stachydrine	C_7_H_13_NO_2_	[M + H]^+^	−0.2	[22]
P119	Phenylalanine	C_9_H_11_NO_2_	[M + H]^+^	0.6	[23]
P215	2-(Acetylamino)-2,6-dideoxy-α-L-galactose	C_8_H_15_NO_5_	[M + H]^+^	0.6	[24]
P296	4-[(5-Hydroxy-3-methyl-1-oxo-2penten-1-yl) amino]-butanoic acid methyl ester	C_11_H_19_NO_4_	[M + H]^+^	−0.4	_
P338	Pinostilbene	C_15_H_14_O_3_	[M + H]^+^	−0.4	[25]
P365	Linamarin	C_10_H_17_NO_6_	[M + H]^+^	−0.2	[26]
P414	3-hydroxy-4′, 5-dimethoxybibenzyl	C_16_H_18_O_3_	[M + H]^+^	−0.4	[27]
P465	Adenosine	C_10_H_13_N_5_O_4_	[M + H]^+^	0.5	[28]
P483 *	Naringenin	C_15_H_12_O_5_	[M + H]^+^	0	[29,30]
P495	Palmitic acid	C_16_H_32_O_2_	[M + NH_4_]^+^	0.4	[31]
P508	Dendrobin A	C_16_H_18_O_4_	[M + H]^+^	0.5	[32]
P522	Stearidonic acid	C_18_H_28_O_2_	[M + H]^+^	−0.5	[33,34]
P688 *	Vanilloside	C_14_H_18_O_8_	[M + H]^+^	1	[35]
P987	Tetradecanoyl-L-Carnitine	C_21_H_41_NO_4_	[M + H]^+^	−0.1	[36,37]
P998 *	N-(3,4,6-Tri-O-acetyl-β-D-glucopyranosyl) piperidine	C_17_H_27_NO_8_	[M + H]^+^	0.1	_
P1291 *	2-Propen-1-yl-2-(acetylamino)-2-deoxy-3-O-β-D-galactopyranosyl-6-Omethyl-α-D-galactopyranoside	C_18_H_31_NO_11_	[M + H]^+^	0.5	_
P1799	Dendronobiloside A	C_27_H_48_O_12_	[M + H]^+^	−0.9	[38]
P2058	Heytrijumalin I	C_34_H_40_O_11_	[M + H]^+^	0.2	[39]
P2453	Acanthoside D	C_34_H_46_O_18_	[M + H]^+^	−0.5	[40,41]
N45	Shikimic acid	C_7_H_10_O_5_	[M − H]^−^	0.6	[42,43]
N50	D-Galactose	C_6_H_12_O_6_	[M − H]^−^	2.7	[44,45]
N81 *	Moscatin	C_15_H_12_O_3_	[M − H]^−^	1.4	[46,47]
N100 *	Tristin	C_15_H_16_O_4_	[M − H]^−^	0.3	[48,49]
N110	Erianthridin	C_16_H_16_O_4_	[M − H]^−^	1.7	[50,51]
N113 *	Dendrophenol	C_16_H_18_O_4_	[M − H]^−^	1.4	[52,53]
N141	Dendroxine	C_17_H_25_NO_3_	[M − H]^−^	2.1	[54,55]
N229	Pinellic acid	C_18_H_34_O_5_	[M − H]^−^	2.5	[56,57]
N241	D-(+)-Trehalose	C_12_H_22_O_11_	[M − H]^−^	1.7	[58]
N341	Dendroside G	C_21_H_34_O_10_	[M − H]^−^	0.4	[59,60]
N358 *	Dendromoniliside B	C_21_H_32_O_11_	[M − H]^−^	0.3	[61]
N404	Raffinose	C_18_H_32_O_16_	[M − H]^−^	1	[62,63]
N484	Vicenin-2	C_27_H_30_O_15_	[M − H]^−^	0.5	[64,65]
N495 *	Rutin	C_27_H_30_O_16_	[M − H]^−^	1	[66]

* The difference value was the compound before rank 50.

## Data Availability

The data used to support the findings of this study can be made available by the corresponding author upon request.

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
