# Peer review of "Traceability Research on Dendrobium devonianum Based on SWATHtoMRM"

_foods, 2023, doi:10.3390/foods12193608_

Round 1
Reviewer 1 Report
I suggest to include the reference DOI: 10.1155/2017/8647212 since this is dealing with a similar topic
Introduction
please check the sentence"...produces pr nationally..."
please check "Dendrobium dendrobium"
please check the sentence (grammar, order of words) "...and the final characteristic information total obtained is low."
The abbreviations should be explained somewhere in the text: especially SWATHtoMRM
Figure 4: please explain the red "*" in the capture
Figure 5: please increase the size of the numbers and text, this is very difficult to read
Figure 5: in the text the ranges 200-300 and 500-700 are mentioned; with the applied scale this is difficult to identify; I suggest to adopt the scale appropriately and mark the regions mentioned in the text.
Figure 6: please explain the numbers shown in the graph in the capture
Figures 7,8,9: please write the plant names in italic
Figure 10: in the capture 4 compounds are mentioned but 5 are shown; the quality of Figure 10 should be improved!
Table 1: what do you mean by "canges in content"? What is the basis for the change? Do you mean that e.g. P1291 is higher in Dd than in Do?
please format Table 1 with appropriate distances between the columns!
Table 2: please check : the compound P414 is not mentioned in ref. 27
please use consistently greek letters in e.g. N-(3,4,6-Tri-O-acetyl-beta-D-glucopyranosyl)...
References:
please check the consistent writing of upper/lower case letters in the article titles.
this is mentioned in the "comments and suggestions" section
Author Response
Response to Reviewer 1 Comments
Point 1: I suggest to include the reference DOI: 10.1155/2017/8647212 since this is dealing with a similar topic.
Response 1: Dear reviewer, please forgive our negligence in literature review. This study was indeed similar to the research object of reference DOI: 10.1155/2017/8647212. However, this study mainly uses the novel SWATHtoMRM method to provide new information for the identification of Dendrobium devonianum and Dendrobium officinale. The references section has been added.
Point 2: Introduction:
please check the sentence"...produces pr nationally..."
please check "Dendrobium dendrobium"
please check the sentence (grammar, order of words) "...and the final characteristic information total obtained is low."
The abbreviations should be explained somewhere in the text: especially SWATHtoMRM
Response 2: Dear reviewer, it has been revised.
Point 3: Figure 4: please explain the red "*" in the capture
Figure 5: please increase the size of the numbers and text, this is very difficult to read
Figure 5: in the text the ranges 200-300 and 500-700 are mentioned; with the applied scale this is difficult to identify; I suggest to adopt the scale appropriately and mark the regions mentioned in the text.
Response 3: Dear reviewer, it has been revised. On the other hand, in Figure 5, the ranges 200-300 and 500-700 were the ranges where molecular weights are concentrated, which was consistent with the range in the figure. The manuscript only uses language description to increase readability, so we feel that it is not necessary to mark the regions mentioned in the text.
Point 4: Figure 6: please explain the numbers shown in the graph in the capture
Figures 7,8,9: please write the plant names in italic
Figure 10: in the capture 4 compounds are mentioned but 5 are shown; the quality of Figure 10 should be improved!
Response 4: Dear reviewer, it has been revised.
Point 5: Table 1: what do you mean by "canges in content"? What is the basis for the change? Do you mean that e.g. P1291 is higher in Dd than in Do?
please format Table 1 with appropriate distances between the columns!
Response 5: Dear reviewer, The meaning of "changes in content" in the article is indeed that P1291 is higher in Dd than in Do. Table 1 has been revised.
Point 6: Table 2: please check : the compound P414 is not mentioned in ref. 27
please use consistently greek letters in e.g. N-(3,4,6-Tri-O-acetyl-beta-D-glucopyranosyl)...
Response 6: Dear reviewer, please forgive our mistake in proofreading the reference. Ref. 27 has been deleted. The greek letters have been revised.
Point 7: References:
please check the consistent writing of upper/lower case letters in the article titles.
Response 7: Dear reviewer, , it has been revised.

Reviewer 2 Report
The paper concerns the identification of two herbal preparations obtained from Dendrobium devonianum and Dendrobium officinale using TOF and MS-MRM combined with SWATHtoMRM technology. The methodology and description of this part is clear. The second part of the manuscript concerns the enrichment pathways of the identified compounds based on the KEGG database.
This analysis was dedicated to selected compounds identified in both preparations, so it is not dedicated to the differentiation of these compounds. In my opinion, this part should be addressed in another paper dealing with pharmacological aspects of the studied herbal products.
The methodology should be improved:
The subchapter "Data processing and statistical analysis" should be more clearly described. What kind of data was imported. Perhaps a supplementary file should be a solution.
2. Sample preparation and chromatographic conditions should be properly described.
3. MetaboAnalyst should have a reference website.
4. Data transferred to the MetaboAnalyst online site should be added as supplementary files.
5. KEGG Pathway Analysis could be dedicated to another work in which case the title should be properly changed to the content.
Author Response
Response to Reviewer 2 Comments
Point 1: The subchapter "Data processing and statistical analysis" should be more clearly described. What kind of data was imported. Perhaps a supplementary file should be a solution.
Response 1: Dear reviewer, in this study, the peak area of ​​each MRM was imported into relevant software for PCA and other analyses. The corresponding analysis steps are solved during the running of the software. The relevant content has been supplemented in the manuscript. We do not think it was necessary to add it in the supplementary.
Point 2: Sample preparation and chromatographic conditions should be properly described.
Response 2: Dear reviewer, it has been revised.
Point 3: MetaboAnalyst should have a reference website.
Response 3: Dear reviewer, it has been revised.
Point 4: Data transferred to the MetaboAnalyst online site should be added as supplementary files.
Response 4: Dear reviewer, data transferred were supplementary S1 and S2, already uploaded in the submission system.
Point 5: KEGG Pathway Analysis could be dedicated to another work in which case the title should be properly changed to the content.
Response 5: Dear reviewer, in this study, KEGG analysis was mainly for a brief analysis of the obtained compounds with different characteristics. The main purpose was to provide a reference for future pathway analysis. It accounts for a small proportion in this study, so we believe that the title does not need to be changed. If you do need to change the title, please let us know in time and we will be happy to change it.